# *Leucobacter manosquensis* sp. nov.—A Novel Bacterial Species Isolated from Healthy Human Skin

**DOI:** 10.3390/microorganisms11102535

**Published:** 2023-10-11

**Authors:** Manon Boxberger, Sibylle Magnien, Angeline Antezack, Clara Rolland, Marine Makoa Meng, Cheikh Ibrahima Lo, Bernard La Scola, Nadim Cassir

**Affiliations:** 1IHU Méditerranée Infection, 19-21 Boulevard Jean Moulin, 13005 Marseille, France; manon.boxberger@hotmail.fr (M.B.); sibylle.magnien@gmail.com (S.M.);; 2Institut de Recherche Pour le Développement (IRD), Assistance Publique-Hôpitaux de Marseille (AP-HM), MEPHI, Aix-Marseille Université, 19 Boulevard Jean Moulin, 13005 Marseille, France; 3École de Médecine Dentaire, Faculté des Sciences Médicales et Paramédicales, Aix-Marseille Université, Boulevard Jean Moulin, 13385 Marseille, France; 4Assistance Publique-Hôpitaux de Marseille (AP-HM), Hôpital Timone, Service de Parodontologie, 264, Rue Saint Pierre, 13385 Marseille, France

**Keywords:** bacteria, taxonogenomics, culturomics, human skin microbiota, new species, *Leucobacter*

## Abstract

Extending our knowledge on human skin microbiota is a challenge to better decipher its role in health and disease. Using the culturomics method, we isolated strain Marseille-Q4368 from the healthy forehead of a 59-year-old woman. We describe here the main characteristics of this bacterium using a taxonogenomic approach. This new bacterial species is Gram-positive, non-motile, and non-spore-forming. Its 16S rRNA sequence exhibited a similarity of 99.59% with *Leucobacter chromiiresistens*, the most closely related species in terms of nomenclature. However, a digital DNA–DNA hybridization analysis between these two species revealed a maximum identity similarity of only 27.5%. We found phenotypical and genomic differences between strain Marseille-Q4368 and its closely related species. These findings underscore the classification of this bacterium as a distinct species. Hence, we propose the name *Leucobacter manosquensis* sp. nov. strain Marseille-Q4368 (=CSUR Q4368 = DSM 112403) for this newly identified bacterial species.

## 1. Introduction

Pharmaceutical and cosmetic innovation have long been based on the search for innovative compounds, often extracted from plants. A major turning point in the history of medicine, with a strong impact on human health, came with the discovery of antibiotics and their natural producers: microorganisms. Fungal and bacterial microorganisms represent promising sources of new compounds for prophylactic or therapeutic treatments. While the medical applications arising from our knowledge on digestive microbiota are now widely used and recognized, they are not yet as well developed concerning cutaneous microbiota and the potential beneficial impact of pre-probiotics and bacteriotherapy on the skin. Not-yet-cultured microorganisms are certainly an underestimated source of interesting compounds. Extending our knowledge on human skin microbiota is a challenge to better decipher its role in health and disease. Recent studies have shown that human skin harbors a wide diversity of microbes, far beyond commonly isolated bacteria. Many of the bacteria that are part of the skin microbiota are difficult to isolate because of difficulties in reproducing the human skin environment in vitro [1]. Knowledge about the skin microbiota has grown exponentially owing in part to the recent advent of metagenomics and 16S rRNA pyrosequencing as tools for microbiological identification. However, culture methods remain an essential tool for studying the characteristics of microorganisms in vivo.

Since 2012, our laboratory has set up an innovative approach aiming at surpassing the limitations associated with culture-independent techniques. These new strategies, called culturomics and taxonogenomics [2,3], consist of combining several isolation conditions to improve culture sensitivity, and MALDI-TOF mass spectrometry [4] and 16SrRNA gene amplification and sequencing [5] for the identification of a large number of isolated bacteria. In particular, the supplementation of natural compounds and antibiotics in the culture medium allowed us to unhide the minority species or to mimic a natural environment through selection processes [6,7]. This technique has allowed the discovery of more than 700 new bacterial species from various environments since its development [8]. As part of the culturomics project aimed at describing the skin microbiota, we isolated the strain Marseille-Q4368 [1]. The *Leucobacter* genus belongs to the *Microbacteriaceae* family and was first proposed in 1996 by Takeuchi et al., with the first species named *Leucobacter komagatae* [9]. *Leucobacter* takes its name from the morphology of the cells, which are rods and light-colored. Currently, there are 41 known species, of which 33 have been validly published with a correct name (https://lpsn.dsmz.de/genus/leucobacter, accessed on 4 September 2023). Moreover, the species found within the genus have been isolated from different environments and *Leucobacter manosquensis* is the first species of this genus to have been isolated from healthy human skin.

Here, we describe *Leucobacter manosquensis*, a new bacterial species isolated from the healthy skin using a taxonogenomic polyphasic approach. It includes phylogenomic analyses, wall fatty acid composition, and phenotypic characterization.

## 2. Materials and Methods

### 2.1. Sample Acquisition and Strain Isolation

Samples were obtained by swabbing a 10 cm^2^ area of skin from the forehead and hands of volunteer healthy women. The samples were collected using sterile swabs soaked in the Culture Top^®^ transport medium (C-top Ae-Ana, Eurobio, Les Ulis, France). The study was validated by the ethics committee Sud-Est IV (reference ID-RCB: 2019-A01508-49). Each sample mixed with the transport media was serially diluted, and 50 µL of each dilution was incubated under aerobic conditions at 31 °C after being seeded (DS) in Columbia Agar (bioMérieux, Marcy l’Etoile, France). To identify the strains, a Matrix-assisted laser desorption ionization time-of-flight (MALDI-TOF) mass spectrometry protein analysis was carried out in triplicate using a Microflex spectrometer (Bruker Daltonics, Bremen, Germany). Whole genome sequencing (WGS) was performed as described below. The strain spectra were imported into the MALDI BioTyper software (version 3.0, Bruker, Bremen, Germany) and analyzed using standard pattern matching with the default parameters. Our database (https://www.mediterranee-infection.com/acces-ressources/base-de-donnees/urms-data-base/, accessed on 4 September 2023) was then incremented with the spectra of this new bacterial species.

### 2.2. Phenotypic Tests

Different growth temperatures (20 °C; 31 °C; 37 °C; 45 °C; and 56 °C), atmosphere conditions (anaerobic, aerobic, and microaerophilic) using generator bags (CampyGEN, Oxoid, Columbia, MD, USA), and pH conditions (5; 6.5; 7.5; 8.5) were tested. The biochemical properties of these strains were tested using API ZYM, API 20 NE, and API 50 CH strips (bioMérieux, Marcy L’Etoile, France) in accordance with the manufacturer’s instructions. To evaluate the bacterial structure, a colony was collected from agar and immersed into a 2.5% glutaraldehyde fixative solution. The suspension was vortexed, passed ten times through a 21-gauge needle to separate the bacterial colonies, and fixed on an uncoated glass slide using cytocentrifugation. A 1% ammonium molybdate negative stain was applied for 1 min before gently washing the slide with 0.2 µm filtered distilled water. The slide was air-dried and examined using scanning electron microscopy on a TM4000 microscope (Hitachi High-Tech, HHT, Tokyo, Japan) with a 15 kV voltage. A motility test was performed using the semi-solid TTC media.

Cellular fatty acid methyl esters (FAME) analysis was performed using GC/MS. Two samples of each strain were prepared with approximately 110 mg of bacterial biomass per tube harvested from several culture plates. The fatty acid methyl esters were prepared as described by Sasser (2006) [10]. GC/MS analyses were carried out as described before [11]. Briefly, the fatty acid methyl esters were separated using an Elite 5-MS column and monitored using mass spectrometry (Clarus 500—SQ 8 S, Perkin Elmer, Courtaboeuf, France). A spectral database search was performed using MS Search 2.0 on the Standard Reference Database 1A (NIST, Gaithersburg, USA) and the FAME mass spectral database (Wiley, Chichester, UK). To evaluate the bacterial structure, a colony was collected from agar and immersed into a 2.5% glutaraldehyde fixative solution. The suspension was vortexed, passed ten times through a 21-gauge needle to separate the bacterial colonies, and fixed on an uncoated glass slide using cytocentrifugation. A 1% ammonium molybdate negative stain was applied for 1 min before gently washing the slide with 0.2 µm filtered distilled water. The slide was air-dried and examined using scanning electron microscopy on a TM4000 microscope (Hitachi High-Tech, HHT, Tokyo, Japan) with a 15 kV voltage. A motility test was performed using the semi-solid TTC media as described by Tittsler et al. [12]. The sporulation was evaluated by collecting a colony from agar in 1 mL of filtered phosphate buffer saline (PBS) and using a thermal shock for 20 min at 60 °C. The biochemical properties of the strain Marseille-Q4368 were tested using API ZYM, API 20 NE, and API 50 CH strips (bioMérieux, Marcy L’Etoile, France) in accordance with the manufacturer’s instructions.

### 2.3. Genome Sequencing, Annotation, and Genome Comparison

The genomic DNA (gDNA) of the strain Marseille-Q4368 was extracted in two steps: a mechanical treatment was first performed with glass beads being acid-washed (G4649-500 g Sigma, St. Louis, MO, USA) using a FastPrep-24™ 5G grinder (mpBio, Santa Ana, CA, USA) at maximum speed (6.5) for 90 s. Then, after 30 min of lysozyme incubation at 37 °C, the DNA was extracted using the EZ1 biorobot (Qiagen, Hilden, Germany) with the EZ1 DNA tissue kit. The elution volume was of 50 µL. The gDNA was quantified using a Qubit assay with a high-sensitivity kit (Life Technologies, Carlsbad, CA, USA) to 0.2 ng/µL. The genomic DNA was then sequenced using the MiSeq technology (Illumina Inc., San Diego, CA, USA) with the paired end strategy prepared with the Nextera XT DNA sample prep kit (Illumina, San Diego, CA, USA). To prepare the paired end library, dilution was performed to require 1 ng of each genome as input. The “tagmentation” step fragmented and tagged the DNA. Then, limited-cycle PCR amplification (12 cycles) completed the tag adapters and introduced dual-index bar codes. After purification with AMPure XP beads (Beckman Coulter Inc., Fullerton, CA, USA), the libraries were then normalized to specific beads according to the Nextera XT protocol (Illumina). The normalized libraries were pooled into a single library for sequencing on the MiSeq. The pooled single-strand library was loaded onto the reagent cartridge and then onto the instrument, along with the flow cell. To improve the quality of the assemblies, an Oxford Nanopore approach was performed on 1D genomic DNA sequencing for the MinIon device using the SQK-LSK109 kit. The library was constructed from 1 µg of genomic DNA without fragmentation and end repair. Adapters were ligated to both ends of the genomic DNA. After purification with the AMPure XP beads (Beckman Coulter Inc., Fullerton, CA, USA), the library was quantified using a Qubit assay with the high-sensitivity kit (Life Technologies, Carlsbad, CA, USA). For bacterial identification, we used the “What’s in my pot” (WIMP) workflow (Epi2me, Oxford Nanopore technologies), which uses the RefSeq sequence database from the NCBI (https://www.ncbi.nlm.nih.gov/refseq/, accessed on 4 September 2023).

The NCBI Prokaryotic Genome Annotation Pipeline was used for genome annotation [13]. The genome sequence data were uploaded to the Type (Strain) Genome Server (TYGS), a free bioinformatics platform available at https://tygs.dsmz.de, for a whole-genome-based taxonomic analysis [14]. Antibiotic resistance genes and the presence of pathogenesis-related proteins were investigated using ABRicate v1.0.1 against ARG-ANNOT [15], EcOH [16], the NCBI Bacterial Antimicrobial Resistance Reference Gene Database [17], PlasmidFinder [18], ResFinder [19], CARD [20], and VFDB [21] using the Online Galaxy platform [22].

Determination of the closest type strain genomes was carried out in two complementary ways. First, all the user genomes were compared against all the type strain genomes available in the TYGS database using the MASH algorithm [23], and the 10 type strains with the smallest MASH distances chosen per user genome. Second, an additional set of 10 closely related type strains was determined via the 16S rDNA gene sequences. These were extracted from the user genomes using RNAmmer [24]. BLAST was then performed with each sequence [25] against the 16S rDNA gene sequence of each of the 12,983 type strains currently available in the TYGS database. This was used as a proxy to find the best 50 matching type strains for each user genome and to calculate the precise distances using the Genome BLAST Distance Phylogeny approach (GBDP) under the algorithm “coverage” and distance formula d5 [26]. These distances were used to determine the 10 closest type strain genomes for each of the user genomes. All pairwise comparisons among the set of genomes were conducted using GBDP and accurate intergenomic distances were inferred using the algorithm “trimming” and distance formula d5. 100 distance replicates were calculated each. The digital DDH (dDDH) values and confidence intervals were calculated using the recommended settings of the GGDC2. In parallel, the degree of genomic similarity between the strains of interest and closely related species was estimated using the OrthoANI software 0.5.0 with the default parameters [27]. The closest species were determined on the basis of DDH. Trees were inferred using FastME 2.1.6.1 [28] from the GBDP distances calculated using the 16S rDNA gene sequences or a whole genome sequence. The branch lengths were scaled in terms of GBDP distance formula d5. The numbers above the branches are the GBDP pseudo-bootstrap support values >60% from 100 replications, with an average branch support of 84.3%. The tree was rooted at the midpoint and regenerated with the iTOL tool v5 [29].

## 3. Results

### 3.1. Identification and Phylogenetic Analysis of Strain Marseille-Q4368

Following a 24 h incubation under aerobic conditions at 31 °C on Columbia agar supplemented with 5% sheep blood, the bacterial colonies from the Marseille-Q4368T strain were subjected to MALDI-TOF MS analysis for identification. The results obtained were inconclusive for identification at the strain level. Then, a phylogenetic analysis was carried out using the 16S rRNA gene sequence. The 16S rRNA sequence of strain Marseille-Q4368 showed 99.5% of similarity when compared to *Leucobacter chromiiresistens* JG 31, the most closely related species in terms of phylogenetic classification (Figure 1). Considering that the obtained percentage similarity of the 16S rRNA gene exceeds the recommended threshold value (98.65%) for defining the species barrier in prokaryotes, additional investigations encompassing morphological, phenotypic, and genomic analyses have been conducted.

### 3.2. Phenotypic Characteristics of Strain Marseille-Q4368

Strain Marseille-Q4368T was facultative aerobic and grew at 31 °C in 24 h. The growth was achieved at a temperature between 20 °C and 37 °C. The optimal pH was between 6.5 and 8.5. NaCl concentrations ranging from 5 g·L^−1^ to 15 g·L^−1^ showed no distinction in growth. The colonies had a translucent gray pigmentation and did not hemolyze. Moreover, the cells were non-motile and did not form spores. This bacterium was Gram-positive (Appendix A). The bacteria were rod-shaped, measuring approximately 1.4 × 0.3 µm. (Figure 2).

For this strain, Marseille-Q4368T, several API galleries were performed, including API ZYM, API 50 CH, and API 20 NE. Positive reactions were observed for the following enzymes: esculin ferric citrate, esterase (C4), esterase lipase (C8), leucine arylamidase, cystine arylamidase, valine arylamidase, naphthol-AS-BI-phosphohydrolase, D-mannitol, D-glucose, D-maltose, malic acid, and α-mannosidase. However, the remaining reactions yielded negative results. Additionally, the catalase test was positive, while the oxidase test was negative. The differential phenotypical characteristics of *Leucobacter manosquensis* strain Marseille-Q4368 and its closest related species are summarized in Table 1. Branched fatty acids were detected in the composition of strain Marseille-Q4368: 12-methyl-tetradecanoic acid (69%), 14-methyl-hexadecanoic acid (16%), and 14-methyl-pentadecanoic acid (7%). Additionally, unsaturated fatty acids were also detected in the cell walls (Table 2).

### 3.3. Comparative Genomic Analysis

The genome size of strain Marseille-Q4368 was 3,184,602 bp long with a 67.34% G + C content. The genome assembly of this strain was achieved on three contigs (with 9.44× coverage). Of the 2976 predicted genes, 2870 were protein-coding genes and 59 were RNAs (3 16S rRNAs, 3 5S rRNAs, 3, 23 rRNAs, 47 tRNAs, and 3 ncRNAs) (Figure 3). 

The in silico resistome of the strain Marseille-Q4368T and the search for the virulence factors of this strain showed neither resistance nor virulence factors. The distribution of the functional classes of the predicted genes according to the clusters of orthologous groups (COGs) of proteins show that the genome of *Leucobacter manosquensis* showed a coherent structure compared to its closely related species (Appendix A).

Genomic data analysis revealed that the highest OrthoANI value was shared between *L. japonicus* and *L. musarum* at 95.3%. On the other hand, the lowest value was shared between *L. japonicus* and *L. aridicollis* at 75.1%. Consequently, strain Marseille-Q4368 had the highest OrthoANI value of 84.3% with *L. chromiiresistens*, which appears to be the closest phylogenetic species (Table 3). Furthermore, strain Marseille-Q4368 had a dDDH value of 27.5% with *L. chromiiresistens*. However, despite the dDDH value of 62.3% obtained between *L. japonicus* and *L. musarum*, the overall DDH values were below the recommended threshold of 70% to define a new prokaryotic species (Table 3) [31].

In terms of genome size, the strain Marseille-Q4368 (3.18 Mbp) had approximately a similar genomic size compared to all the genomes of the *Leucobacter* genus. However, it possessed a smaller number of proteins compared to the other genomes, except for that of *L. massiliensis* (3.14 Mbp). Additionally, the tRNA molecules were more numerous than the rRNA and other RNA molecules, ranging between 46, 47, and 48 depending on the *Leucobacter* species analyzed. Furthermore, the difference in the G + C percentage between the species was very low, mostly being less than 4%. This reinforces our thesis that the Marseille-Q4368 strain is a new species within the *Leucobacter* genus. More statistical details from the genomes are showed in Table 4.

### 3.4. Description of Leucobacter manosquensis sp. nov.

*Leucobacter manosquensis* (ma.nos.quen’sis. N.L. masc. adj. manosquensis, “of Manosque,” referring to the place where M&L Laboratories is located). Strain Marseille-Q4368 exhibited facultative aerobic growth, with a growth rate of 31 °C within 24 h. This particular bacterium was Gram-positive and had non-motile, non-spore-forming bacterial cells that were rod-shaped, measuring approximately 1.4 × 0.3 µm. 12-methyl-tetradecanoic acid (69%), 14-methyl-hexadecanoic acid (16%), and 14-methyl-pentadecanoic acid (7%) were the main branched fatty acids retrieved in strain Marseille-Q4368. Esculin ferric citrate, esterase (C4), esterase lipase (C8), leucine arylamidase, cystine arylamidase, valine arylamidase, naphthol-AS-BI-phosphohydrolase, D-mannitol, D-glucose, D-maltose, malic acid, and α-mannosidase were the present enzymes. It tested positive for a catalase test and negative for an oxidase test.

The genome of strain Marseille-Q4368^T^ was 3.18 Mbp with 67.3 mol% of G + C content.

The genome and 16S rRNA sequence were deposited in GenBank under accession numbers MW583464 and JAFEVO000000000, respectively.

The type strain Marseille-Q4368^T^ (=CSUR Q4368 = DSM 112403) was isolated from the forehead of a 59-year-old woman with healthy skin.

## 4. Conclusions

The skin is the largest organ of the human body, and because of its external nature, skin disorders have been the focus of medical research and cosmetic trials since ancient times. Having evolved together, scientific knowledge and cosmetic formulations are now inseparable. It should be noted that cosmetic and pharmaceutical regulations are particularly demanding, and are constantly updated in light of new scientific advances. The skin is colonized by a large number of organisms with which it evolves in symbiosis: the cutaneous microbiota. The number of symbiotic microorganisms is greater than the number of skin cells. Although it is not (yet) possible under actual European regulations to add living microorganisms to a formulation, growing knowledge on this subject already offers clear new prospects. Thus, the discovery of new bacteria could be a starting point for innovating component findings. Leucobacter manosquensis was discovered from a healthy skin swab sample. Both phylogenetic and phenotypic analyses revealed several different characteristics of the strain Marseille-Q4368 when compared to other members of the genus Leucobacter, which belongs to the Microbacteriaceae family and the Actinobacteria phylum and was first proposed in 1996 [9]. Leucobacter manosquensis strain Q4368 is the first species belonging to the Leucobacter genus that has been isolated from healthy human skin. Indeed, the dDDH and OrthoANI percentages of the strain Marseille-Q4368 are below the admitted threshold percentage of <70% and <95%, respectively, to define a new prokaryotic species [31]. Therefore, we propose Marseille-Q4368 as the type strain of a new species, within the genus Leucobacter, named Leubacter manosquensis. Gr. masc. adj. [λευκός] leukos, clear, light; N.L. masc. n. bacter, rod; N.L. masc. n. Leucobacter, colorless rod. Manosquensis, translitt. L. adj., “of Manosque,” referring to the place where M&L Laboratories (one of the founders) is located.

## Figures and Tables

**Figure 1 microorganisms-11-02535-f001:**
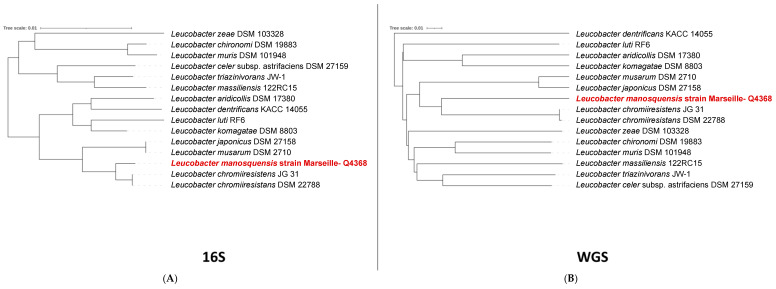
(**A**) 16S rRNA-based phylogenetic tree (**B**) Whole-genome-based phylogenetic tree highlighting the position of *L. manosquensis* sp nov., strain Marseille-Q4368^T^ relative to other closely related bacterial taxa.

**Figure 2 microorganisms-11-02535-f002:**
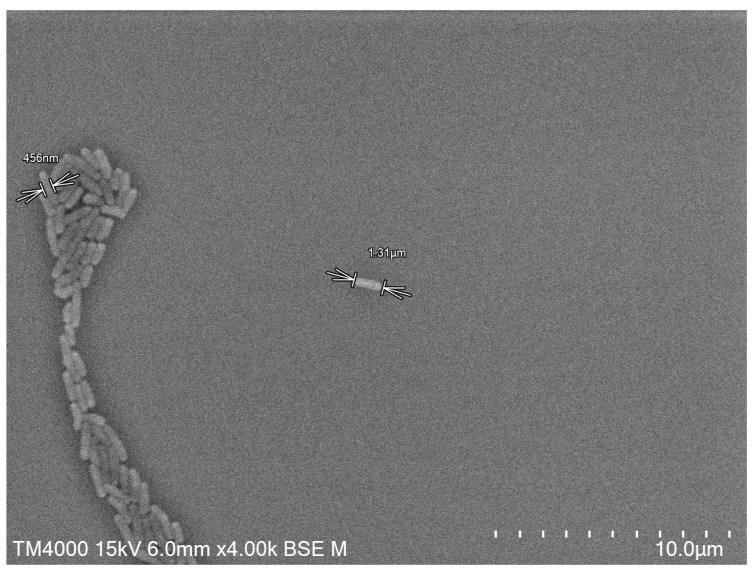
Scanning electron microscopy of *L. manosquensis* sp. nov., strain Marseille-Q4368^T^ using a TM4000 microscope (Hitachi High-Tech, HHT, Tokyo, Japan). The scale bar represents 10 µm.

**Figure 3 microorganisms-11-02535-f003:**
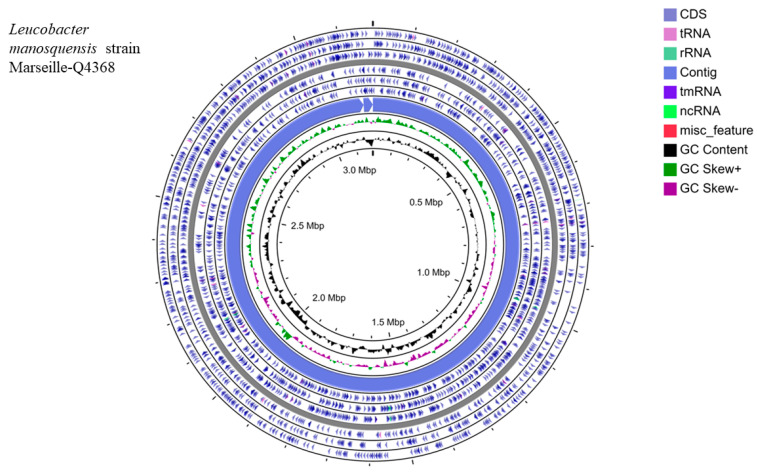
Graphical circular map of the genome from *L. manosquensis* strain Marseille-Q4368^T^ obtained using the CGView Server online tool [30].

**Table 1 microorganisms-11-02535-t001:** Essential phenotypic criteria shared between 1, *Leucobacter manosquensis* strain Marseille-Q4368 and closely phylogenetical species such as 2, *Leucobacter chromiiresistens* JG 31 and 3, *Leucobacter musarum* CBX 152.

Properties	1	2	3
Cell diameter (µm)	1.4 × 0.3	0.3 × 0.7	NA
Oxygen requirement	FA	Aerobic	Aerobic
Gram stain	+	+	+
Motility	−	−	−
Endospore formation	−	−	−
Alkaline phosphatase	−	+/−	NA
Catalase	+	+	+
Oxidase	−	−	−
α-glucosidase	−	−	NA
β-galactosidase	−	−	−
Acid from:			
N-acetylglucosamine	−	−	−
L-arabinose	−	+	−
D-ribose	−	−	−
D-mannose	−	−	−
D-mannitol	+	−	+
D-glucose	+	−	−
D-fructose	−	−	−
D-maltose	+	−	−
D-lactose	−	−	−
G + C content (mol%)	67.34	NA	66.77
Isolation source	Human healthy skin	Uncontaminated soil	Nematode

FA = Facultative anaerobic; + = positive reaction; − = negative reaction; NA = data not available.

**Table 2 microorganisms-11-02535-t002:** Fatty acid component (%) analysis from *Leucobacter manosquensis* Marseille-Q4368 (Lman), *Leucobacter chromiireducens* JG 31 (Lchr), *Leucobacter aridicollis* L-9T (Lari), and *Leucobacter celer* NAL101 (Lcel).

Fatty Acids	Name	Lman	Lchr	Lari	Lcel
C_15:0 anteiso_	12-methyltetradecanoic acid	60.7	55.9	49	46.5
C_17:0 anteiso_	14-methyl-hexadecanoic acid	23.8	23.5	24	21.1
C_16:0 iso_	14-methyl-pentadecanoic acid	13.1	12.2	12	16.0
C_16:0_	Hexadecanoic acid	1.4		11	
C_15:0 iso_	13-methyl-tetradecanoic acid	TR	0.62		9.7

**Table 3 microorganisms-11-02535-t003:** Comparative genomic data including ANI (top right) and dDDH (bottom left) values between *Leucobacter manosquensis* Marseille-Q4368 (Lman) and its closely related species such as *Leucobacter celer* NAL101 (Lcel), *Leucobacter triazinivorans* JW-1 (Ltri), *Leucobacter massiliensis* 122RC15 (Lmas), *Leucobacter zeae* CC-MF41(Lzea), *Leucobacter luti* RF9 (Llut), *Leucobacter chromiiresistens* JG 31 (Lchr), *Leucobacter japonicus* CBX 130 (Ljap), and *Leucobacter musarum* CBX 152 (Lmus).

	Lari	Lcel	Ltri	Lmas	Lzea	Llut	Lchr	Lman	Ljap	Lmus
Lari		76.4%	76.6%	76.4%	76.4%	76.9%	76.5%	76.5%	75.1%	75.0%
Lcel	21.8%		85.7%	81.3%	79.7%	79.1%	79.5%	78.2%	78.2%	77.9%
Ltri	21.9%	30.1%		81.2%	79.8%	78.9%	79.8%	78.3%	77.9%	77.7%
Lmas	21.4%	24.9%	24.2%		78.9%	78.5%	79.3%	77.8%	77.4%	77.2%
Lzea	21.4%	23.3%	22.9%	22.7%		78.4%	79.4%	77.8%	77.6%	77.5%
Llut	21.8%	23.0%	22.9%	22.4%	22.2%		78.8%	77.4%	77.0%	77.0%
Lchr	21.7%	23.6%	23.3%	22.8%	22.8%	22.4%		84.3%	81.1%	81.0%
Lman	20.9%	22.6%	21.8%	22.0%	21.5%	21.5%	27.5%		79.8%	79.7%
Ljap	21.2%	23.3%	22.10	21.6%	21.7%	21.3%	24.0%	22.7%		95.3%
Lmus	21.0%	23.0%	22.2%	21.7%	21.7%	21.2%	23.7%	22.6%	62.3%	

**Table 4 microorganisms-11-02535-t004:** Properties from genome analysis.

Name	RefSeq	Size (Mbp)	G + C %	Protein	rRNA	tRNA	Other RNA	Pseudogene	Total Genes
*Leucobacter massiliensis* 122RC15	MWZD00000000	3.14	71.0	2766	4	46	3	53	2872
*Leucobacter manosquensis* Marseille-Q4368	JAFEVO000000000	3.18	67.3	2776	9	47	3	149	2984
*Leucobacter chromiiresistens* JG 31	AGCW00000000	3.22	70.3	2919	9	48	3	34	3013
*Leucobacter musarum* subsp. *musarum* CBX152	JHBW00000000	3.44	66.8	3096	4	47	3	20	3590
*Leucobacter aridicollis* L-9	QYAE00000000	3.46	67.3	3115	4	47	2	0	3168
*Leucobacter zeae* CC-MF41	QYAB00000000	3.47	70.6	2914	4	47	3	137	3105
*Leucobacter triazinivorans* JW-1	CP035806	3.48	69.4	3048	6	46	3	38	3141
*Leucobacter japonicus* CBX130	JHBX00000000	3.59	66.8	3262	3	47	3	31	3346
*Leucobacter luti* RF6	QYAG00000000	3.62	69.5	3071	5	46	3	32	3156
*Leucobacter celer* subsp. *astrifaciens* CBX151	JHEI00000000	4.14	69.1	3589	3	48	3	43	3686

## Data Availability

Data supporting reported results are available from the corresponding author.

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
