# Peer review of "Leucobacter manosquensis sp. nov.—A Novel Bacterial Species Isolated from Healthy Human Skin"

_microorganisms, 2023, doi:10.3390/microorganisms11102535_

Round 1

Reviewer 1 Report

The article describes the phenotypic, enzymatic and genomic characterization of a new bacterium isolated from healthy human skin. Overall, the analyses are well-performed and open the opportunity to apply similar approaches to discover new organisms, and to further look for this organism in skin metagenomic datasets. Minor suggestions below:

Title: Suggest to change to Description of Leucobacter manosquensis sp. nov. a novel bacterial species isolated from healthy human skin

Line 44: Is this an author's statement? Please include reference.

Line 58: Technique instead of technic?

Line 60: decline instead of declined?

Line 161: Suggest to replace "Each sequence was then BLASTed" to "BLAST was then performed with each sequence.

Can the authors upload a figure 1 with higher resolution? As of now is very hard to read

Did the authors perform a prophage analysis to determine if the organism has any prophage? It would be interesting to know this aspect.

No further comments on the English language

Author Response

Thank You for your comments

Title: Suggest to change to Description of Leucobacter manosquensis sp. nov. a novel bacterial species isolated from healthy human skin

We changed the title

Line 44: Is this an author's statement? Please include reference.

We deleted this sentence

Line 58: Technique instead of technic?

We modified it in the text

Line 60: decline instead of declined?

We modified it in the text

Line 161: Suggest to replace "Each sequence was then BLASTed" to "BLAST was then performed with each sequence.

We modified it it the text

Can the authors upload a figure 1 with higher resolution? As of now is very hard to read

We uploaded a Figure 1 with higher resolution

Did the authors perform a prophage analysis to determine if the organism has any prophage? It would be interesting to know this aspect.

We did not perform a prophage analysis but intend to extend ou analysis of prophages in bacterial skin in a further work

Reviewer 2 Report

Dear Authors, Thank you for your interesting topic. I have several comments that you could consider to correct before publication.

1/ "As part of the culturomics project declined on the exploration of the skin microbiota, we isolated Leucobacter manosquensis strain Marseille-Q4368 [1]". In lines 59-61, it is better if the authors only mention "strain Marseille-Q4368" which is the preliminary data for identifying a new bacterium, Leucobacter manosquensis.

2/ Please add a citation for "The workflow WIMP was chosen for bioinformatic analysis in live".(line 147).

3/ In Figure 1, please use Italic font for species names and use a higher-resolution figure.

4/ "(3 16S rRNA, 3 5S rRNAs, 3 23S rRNAs, 47 tRNAs, and 3 ncRNAs)": please revise to (three 16S rRNAs, three 5S rRNAs, three, 23 rRNAs, 47 tRNAs, and three ncRNAs) for easier follow.

5/ In Table 3 and Table 4, please highlight the data of L. manosquensis so that the readers can easily focus on the differences of L. manosquensis to other related species.

Author Response

Thank You for your comments

1/ "As part of the culturomics project declined on the exploration of the skin microbiota, we isolated Leucobacter manosquensis strain Marseille-Q4368 [1]". In lines 59-61, it is better if the authors only mention "strain Marseille-Q4368" which is the preliminary data for identifying a new bacterium, Leucobacter manosquensis.

Accordingly, we modified it in the text

2/ Please add a citation for "The workflow WIMP was chosen for bioinformatic analysis in live".(line 147).

We added references for the WIMP workflow

3/ In Figure 1, please use Italic font for species names and use a higher-resolution figure.

Accordingly, we modified the Figure 1

4/ "(3 16S rRNA, 3 5S rRNAs, 3 23S rRNAs, 47 tRNAs, and 3 ncRNAs)": please revise to (three 16S rRNAs, three 5S rRNAs, three, 23 rRNAs, 47 tRNAs, and three ncRNAs) for easier follow.

We modified it in the text

5/ In Table 3 and Table 4, please highlight the data of L. manosquensis so that the readers can easily focus on the differences of L. manosquensis to other related species.

Accordingly, we modified the tables 3 and 4

Reviewer 3 Report

I have reviewed the manuscript titled "Extending our knowledge on human skin microbiota: Discovery of Leucobacter manosquensis sp. nov. strain Marseille-Q4368." 

The study presented in this manuscript addresses a critical challenge in the field of microbiology, namely, the exploration of human skin microbiota and its role in health and disease. The use of the culturomics method to isolate a new bacterial species, Leucobacter manosquensis sp. nov. strain Marseille-Q4368, from the healthy forehead of a 59-year-old woman is a significant contribution to the field. The authors have employed a taxonogenomic approach to thoroughly characterize this novel bacterium. The description of its main characteristics, such as being gram-positive, non-motile, and non-spore-forming, is well-documented and adds valuable information to our understanding of bacterial diversity. The discovery that its 16S rRNA sequence exhibits a high similarity with Leucobacter chromiiresistens but with a digital DNA-DNA hybridization analysis revealing significant genomic differences is particularly noteworthy. This highlights the importance of genomic data in bacterial classification and taxonomy. Furthermore, the proposal of the name "Leucobacter manosquensis sp. nov. strain Marseille-Q4368" for this newly identified bacterial species is appropriate and follows established nomenclature guidelines. The inclusion of reference strains (CSUR Q4368 and DSM 112403) further enhances the credibility and utility of this discovery for the scientific community. In summary, this manuscript represents a commendable effort to expand our knowledge of the human skin microbiota, and it significantly contributes to the field of microbiology by describing a new bacterial species. The methodology is sound, the results are robust, and the findings are of broad scientific interest. I recommend the publication of this work in your journal.

Author Response

Thank You for your valuable comments